# Nitrogen-Doped Hierarchical Porous Carbon Derived from Coal for High-Performance Supercapacitor

**DOI:** 10.3390/molecules28093660

**Published:** 2023-04-23

**Authors:** Leiming Cai, Yanzhe Zhang, Rui Ma, Xia Feng, Lihua Yan, Dianzeng Jia, Mengjiao Xu, Lili Ai, Nannan Guo, Luxiang Wang

**Affiliations:** State Key Laboratory of Chemistry and Utilization of Carbon Based Energy Resources, College of Chemistry, Xinjiang University, Urumqi 830017, China; clm19961107@163.com (L.C.);

**Keywords:** herarchical porous carbon, pore engineering, nitrogen doping, coal, supercapacitors

## Abstract

The surface properties and the hierarchical pore structure of carbon materials are important for their actual application in supercapacitors. It is important to pursue an integrated approach that is both easy and cost-effective but also challenging. Herein, coal-based hierarchical porous carbon with nitrogen doping was prepared by a simple dual template strategy using coal as the carbon precursor. The hierarchical pores were controlled by incorporating different target templates. Thanks to high conductivity, large electrochemically active surface area (483 m^2^ g^−1^), hierarchical porousness with appropriate micro-/mesoporous channels, and high surface nitrogen content (5.34%), the resulting porous carbon exhibits a high specific capacitance in a three-electrode system using KOH electrolytes, reaching 302 F g^−1^ at 1 A g^−1^ and 230 F g^−1^ at 50 A g^−1^ with a retention rate of 76%. At 250 W kg^−1^, the symmetrical supercapacitor assembled at 6 M KOH shows a high energy density of 8.3 Wh kg^−1,^ and the stability of the cycling is smooth. The energy density of the symmetric supercapacitor assembled under ionic liquids was further increased to 48.3 Wh kg^−1^ with a power output of 750 W kg^−1^ when the operating voltage was increased to 3 V. This work expands the application of coal-based carbon materials in capacitive energy storage.

## 1. Introduction

As the energy crisis intensifies and fossil resources dwindle, there is an imperative to develop new types of energy storage devices, the most common of which are lithium-ion batteries and supercapacitors [1,2,3]. Supercapacitors offer the advantages of high-power density, rapid charge accumulation/release, and long cycle times compared to batteries, but low energy density, which also severely limits their practical application [4,5,6,7]. In general, electrode materials are critical to achieving high-performance supercapacitors. Among the supercapacitor electrode materials, pseudocapacitance materials (RuO, MnO_2_, Ni(OH)_2_, et al.) display ultrahigh specific capacitance, but their poor electronic conductivity leads to higher electrode resistance and lower power density compared with EDLC (double electric layer capacitor) materials. Recently, two-dimensional MXene materials have shown high capacitance, power density, and energy density, but the synthesis process is more complex [8,9,10]. The most widely used electrode material in supercapacitors is carbon material [11,12,13,14], which is widely available, cheap, easy to obtain, has good electrical conductivity, easy surface modification, large specific surface area, and adjustable morphology [15,16,17].

Hierarchical porous carbon materials with many pores and a reasonable ratio of different types of pores provide fast ion transport pathways, while the large surface area accessible to ions leads to excellent rate performance, high capacitance, and high energy density [18,19,20,21]. Micropores can act as active sites for ion storage, which is an important condition for determining high specific capacitance [22,23]. Mesopores may provide transport channels for ions from the electrolyte, while macropores may act as ion buffer regions [24,25]. This is critical for superfast supercapacitors designed to operate at high charge/discharge rates. For this purpose, hierarchically porous carbon materials with reasonably interconnected hierarchical pores are highly desirable. To date, breakthroughs have been made in the preparation of carbon materials with hierarchical porous structures. However, it usually involves complex and environmentally unfriendly fabrication processes [26,27,28,29,30,31]. Thus, facile and green strategies for synthesizing hierarchical porous carbon still face enormous challenges.

Excitingly, it is generally accepted that the surface properties of carbon, such as surface wettability, electron conductivity, and electrochemical reactivity, can be modulated by doping the carbon matrix with other heteroatoms (e.g., N, O, S, and B) to further improve the capacitive properties [32,33,34,35,36,37]. Doping nitrogen into carbon is the most promising approach to improving capacitance. Accordingly, the nitrogen-doped porous carbons are always prepared from activating nitrogen-containing precursors or post-treatment with a nitrogen-containing agent, which exhibits excellent rate and capacitive performance for supercapacitors. However, nitrogen-doped carbons obtained in the above manners are often unable to have multiple properties of hierarchical porous structure, high surface nitrogen content, and large specific surface area, which greatly limits their large-scale practical applications [38,39]. Therefore, an integrated design through morphological modifications, different pore ratios, and heteroatom introduction should be a promising strategy to advance supercapacitor carbon electrodes.

Herein, we have developed a practical and environmentally friendly method for the synthesis of nitrogen-doped hierarchical porous carbon (h-CPC) materials with mixed nanosheet and bulk structures through low-cost carbonization of coal. During the decomposition of potassium citrate and g-C_3_N_4_, the escape of gas induces the generation of porous structures. The generated K compounds can be used as activated agents to further develop the porous structures of the prepared samples. Moreover, the g-C_3_N_4_ can be acted as a nitrogen source to introduce nitrogen in the carbon skeletons. Therefore, the prepared h-CPC has a high conductivity network, a large surface area accessible to ions, and hierarchical porosity, which contributes to faster dynamic ion transport and adsorption [40]. As the electrode material for KOH solution in a three-electrode system, h-CPC shows a high capacitance of 302 F g^−1^ at 1 A g^−1^ and an excellent capacity retention of 76% at 50 A g^−1^. At 250 W kg^−1^, the assembled symmetrical supercapacitor reveals a specific energy density of 8.3 Wh kg^−1^. The h-CPC electrode was also tested on 1 M Na_2_SO_4_ and ionic liquids, which also shows excellent electrochemical performance. The results of the above electrochemical tests show that it is a promising electrode material for supercapacitors with high performance and high-capacity retention.

## 2. Results and Discussion

### 2.1. Schematic Diagram

The preparation process of all samples is shown in Figure 1. Hierarchical porous carbon materials with a combination of heterogeneous nanosheets and irregular porous particles were synthesized using a feasible dual template strategy. The formation of h-CPC consists of two main sections. In the low-temperature phase (<650 °C), the coal and potassium citrate undergo decomposition. Potassium citrate decomposes into potassium carbonate. At higher temperatures, potassium carbonate decomposes into CO_2_ and K_2_O. Subsequently, CO_2_ and K_2_O further react with the carbon skeleton to obtain a porous structure, which is essential to produce microporosity. In addition, gases from the decomposition process contribute to the formation of porous structures [38,41].
2K_3_C_6_H_5_O_7_ → 3K_2_CO_3_ + 9C + 5H_2_O
K_2_CO_3_ → CO_2_ + K_2_O
CO_2_ + C → 2CO
K_2_O + C → 2K + CO

In the high-temperature stage (>650 °C), g-C_3_N_4_ begins to decompose and produces more small nitrogen-containing molecules, which can fully react with active surface atoms and introduce abundant nitrogen species into the carbon skeleton [42]. In addition, potassium citrate and g-C_3_N_4_ can also be used as carbon sources, thus increasing sample yield. Hence, the yield of h-CPC can reach 0.35 g, which is much higher than that of microporous coal-based porous carbon (Mi-CPC, 0.15 g) and mesoporous coal-based porous carbon (Me-CPC, 0.2 g) in Appendix A. More interestingly, g-C_3_N_4_ plays an important role not only in tuning the pore structure but also in changing the surface composition, which is essential for improving the energy storage capacity.

### 2.2. Shape of the Sample

Scanning electron microscopy (SEM) and transmission electron microscopy (TEM) were used to observe the morphology and microstructure. As seen in Figure 2a,d, Mi-CPC exhibits a sheet-like structure. In contrast, Me-CPC shows an irregular three-dimensional (3D) hierarchical structure in Figure 2c,f, mainly due to persistent gas release through pyrolysis of the g-C_3_N_4_ template. Different from the single template, dual templates of g-C_3_N_4_ and potassium citrate can endow different morphologies to the products. As shown in Figure 2b,e, the morphology of h-CPC with both nanosheet and 3D hierarchical structure, which is the combined action of potassium citrate and g-C_3_N_4_. As seen in the TEM images of h-CPC (Figure 2g–i), the carbon layer and the nanopores distributed in it can be clearly seen.

### 2.3. Surface Properties and Pore Structure Analysis

Two major carbon peaks of 24° and 42° were shown in the X-ray diffraction (XRD) spectra (Figure 3a), indicating the amorphous structural characteristics of all samples. Lower and wider peaks (002) at 24° indicate a lower degree of graphitization of Mi-CPC than Me-CPC and h-CPC. This can be attributed to the increased activation of potassium citrate compared to g-C_3_N_4_. In addition, (002) peaks of h-CPC and Me-CPC are shifted to the right compared to Mi-CPC, indicating a reduction in the overall crystallographic spacing caused by doped heteroatoms. D and G bands in the Raman spectroscopy are typically used to assess the degree of graphitization of carbon materials. The degree of graphitization is assessed by the I_D_/I_G_ intensity ratio in bands D and G, with the D band at 1338 cm^−^^1^ representing the disordered carbon structure and the G band at 1590 cm^−^^1^ representing the graphitized structure [43]. This demonstrates that the addition of g-C_3_N_4_ results in an increase in I_D_/I_G_ and a decrease in graphitization, which is consistent with the XRD pattern (Appendix A).

N_2_ adsorption/desorption isotherm samples were measured to investigate the porous structure of the samples. As shown in Figure 3b, Mi-CPC shows a Type I profile, indicating that potassium citrate is primarily used to generate microporous structures [44]. The corresponding pore size distribution further confirms the dominant micropores, which are activated by K_2_CO_3_ generated by potassium citrate during the heated process. At high temperatures, K_2_CO_3_ will further decompose to form CO_2_ and K_2_O, which further etch the carbon skeleton to produce microporosity [45,46]. Me-CPC activated only by g-C_3_N_4_ exhibits typical mesoporous dominant features [47]. Notably, strong adsorption at P/P_0_ < 0.01 and a hysteresis loop at 0.4 < P/P_0_ < 0.1 can be clearly seen in Figure 3b, which is typical of type IV isotherms, indicating the coexistence of micropores and mesopores in h-CPC prepared using the dual-template approach. Additionally, the distribution of pore sizes in Figure 3c further confirms this conclusion. The larger the adsorption capacity, the higher the specific surface area. Thus, h-CPC has the highest specific surface area of 1735 m^2^ g^−^^1^, which is much higher than Mi-CPC (1268 m^2^ g^−^^1^) and Me-CPC (320 m^2^ g^−^^1^, Appendix A), suggesting that the activation effects of potassium citrate and g-C_3_N_4_ may be complementary. Micropores are active centers for ion adsorption, and mesopores provide fast ion transport channels [19,21,24]. These results suggest that the specific surface area and pore structure can be tuned by controlling different templates, which is beneficial for ion storage and rapid ion transport.

Surface-doped heteroatoms can modify the electrochemical properties of porous carbon materials [48]. The presence of surface N and O elements in h-CPC has also been confirmed by X-ray photoelectron spectroscopy (XPS). The percentage of N atoms increased from 1.40% (Mi-CPC) to 18.31% (h-CPC) when g-C_3_N_4_ was added, as shown in Appendix A. However, too much nitrogen doping can damage the structure of the carbon skeleton, which in turn is not conducive to rapid electron transport. It is reasonable to assume that the surface nitrogen content of h-CPC at 18.31% is sufficient to improve the performance of the carbon material. As seen in the high-resolution C1s spectra (Figure 3d), four distinct characteristic peaks were located at a frequency of 284.8, 286.1, 288.2, and 288.8 eV, representing the keys C-C=C, C-N, C-O, and C=O, respectively. N 1s (Figure 3e) spectra can be fitted to three different peaks, including pyridine-N (397.7 eV), pyrrolic N (399.9 eV), and quaternary nitrogen (402.5 eV), respectively. O1s peaks (Figure 3f) are decomposed into four peaks that correspond to C=O, C-OH, C-O-C, and -COOH at 531.1, 532.6, 532.9, and 535.3 eV, respectively. The surface O and pyridine N could be used as Faraday reaction sites to introduce additional pseudocapacitance, and quaternary ammonium nitrogen could transfer carbon lattices electronically. More importantly, doping of N and O can significantly improve the surface wettability of carbon materials and promote the adsorption of electrolyte ions.

### 2.4. Electrochemical Properties of All Samples in a Three-Electrode System

Given its abundant surface N content, large specific surface area, and hierarchical pore structure, it is expected that h-CPC will be an ideal electrode material for use in supercapacitors. To verify the electrochemical properties of the resulting samples, galvanostatic charge/discharge (GCD) and cyclic voltammetry (CV) measurements were performed in a three-electrode system using 6 M KOH as the electrolyte. The GCD plots show a slightly deflected isosceles triangle (Figure 4a), and the CV plots show a rectangular shape (Figure 4b), showing the combination of EDLC and pseudocapacitive storage behaviors of all the samples. Compared to the comparison sample, h-CPC has the longest discharge time and the largest current response, demonstrating that it has the best capacitive performance. The relationship between capacitance and current density is shown in Figure 4c. At current density 1–50 A g^−^^1^, h-CPC displays significantly higher specific capacitances than Mi-CPC and Me-CPC. Specifically, the h-CPC electrode has the highest specific capacitance of 302 F g^−^^1^ at 1 A g^−^^1^ and capacity retention of 76.1% at 50 A g^−^^1^ (Figure 4c), which is much better than most reported carbon materials (Appendix A). This excellent capacitive performance is mainly due to the hierarchical structure of h-CPC. In general, a large pore volume provides more active sites, which can be further evaluated by cyclic voltammetry of electrochemically active surface area (ECSA, Figure 4d). The ECSA of h-CPC (483 m^2^ g^−^^1^) was higher than that of Mi-CPC (383 m^2^ g^−^^1^) and Me-CPC (267 m^2^ g^−^^1^), further indicating that h-CPC has the highest ECSA.

The contributions of pseudocapacitance and EDLC for all samples were calculated by Equations (1) and (2).
*i* = *kν^b^*(1)
*log*(*i*) = *b log*(*v*) + *log*(*k*)(2)

This value of *b* is in the range of 0.5 to 1 and is commonly used to assess the kinetics of redox reactions. The value of *b* approaching 1 represents EDLC and is not affected by diffusion-control processes and charge-transfer limitations [49]. Additionally, as the *b* value approaches 0.5, it corresponds to the internal Faraday pseudocapacitance and gradually decreases as the scan rate increases [50]. The *b* value of h-CPC is in the range of 0.85 to 0.95, and this function also exhibits a progressive upward relationship (Figure 4e). These results confirm the ideal capacitance feature with ultrafast response kinetics. Figure 4f shows that the fitted results for h-CPC at 500 mV s^−^^1^ show a contribution from rapid response kinetics up to 92.9%. Interestingly, the rapid response kinetics of the h-CPC electrode remained relatively stable with increasing scan rates (Figure 4g). When the current density is high, the GCD curve exhibits a slightly deflected triangular shape (Figure 4h), while the CV plot shows a rectangle with high sweep rates (Figure 4i). Figure 4j shows the Nyquist plot of the EIS test in the three-electrode system. Compared to Mi-CPC and Mi-CPC, h-CPC has a smaller radius of the semicircle in the mid-frequency region, and the slope of the high-frequency region is almost vertical, indicating its low charge transfer resistance and effective ion diffusion in the hierarchical porous structure. The retention capacity of the h-CPC device is 100% after 10,000 cycles at 5 A g^−^^1^ (Figure 4k). The results show that the retention capacitance is 100% after 10,000 cycles, indicating significant cyclic stability of the h-CPC electrodes. The correlation analysis above indicates that the integration of abundant micropores, mesopores, hierarchical porous structure, and rich surface heteroatoms effectively ensures sufficient active sites, fast charge, and ion transport capabilities, leading to excellent capacitive and kinetics performance.

### 2.5. Electrochemical Performance of h-CPC in a 6 M KOH Electrolyte for a Two-Electrode System

To assess the practical potential of h-CPC, symmetrical supercapacitors were assembled in 6 M KOH electrolytes. Even at 50 A g^−^^1^, the GCD profiles still maintain a triangular shape (Figure 5a), and the CV curve maintains a nearly rectangular shape with no obvious deformation even at scan rates up to 500 mV s^−^^1^ (Figure 5b), indicating exceptional cycle reversibility and rapid charging and discharging capabilities. The symmetrical h-CPC supercapacitor has excellent capacitance and rate performance.

A specific capacitance of 240 F g^−^^1^ was achieved at 1 A g^−^^1^ and 180 F g^−^^1^ at 50 A g^−^^1^ with a capacitance retention rate of 75% (Figure 5c). The maximum energy density of the device was calculated to be 8.3 Wh kg^−^^1^ at 250 W kg^−^^1^ and still reached 6.6 Wh kg^−^^1^ at 12,500 W kg^−^^1^ (Figure 5d), which is significantly better than the carbonaceous material already reported. The assembled devices also exhibited 100% Coulomb efficiency and 88% capacitance retention after 10,000 cycles (Figure 5e), respectively. As can be seen from Figure 5e, after 10,000 cycles, the GCD changes minimally and demonstrates high reversibility and cycling stability of the h-CPC electrode. To extend the application of h-CPC-based supercapacitors, we assembled supercapacitors with two electrodes at 1 M Na_2_SO_4_. The h-CPC is capable of operating at a high voltage of 1.6 V and exhibits remarkable capacitance, rate performance, and cycling stability (Appendix A).

### 2.6. Electrochemical Performance of h-CPC in a Two-Electrode System with 1-Ethyl-3-Methylimidazole Tetrafluoroborate (EMIM BF_4_) Electrolyte

Finally, the h-CPC supercapacitor was assembled into the EMIM BF_4_ electrolyte to extend the working voltage. The GCD profiles exhibit triangular isosceles and approximately linear time-potential features (Figure 6a), demonstrating EDLC dominance and excellent kinetic reversibility of h-CPC at a high operating voltage of 3 V. CV curves maintain a quasi-rectangular shape at different scan rates ranging from 5 to 500 mV s^−^^1^, further demonstrating ultra-fast electrochemical response and excellent cycle reversibility (Figure 6b). At a current density of 1 A g^−^^1^, the device reaches a specific capacitance of 155 F g^−^^1^ (Figure 6c). It is worth noting that the specific capacitance remains at 50 F g^−^^1^ as the current density increase to 50 A g^−^^1^ with a retention of 32% (Figure 6c). Due to the extended operating voltage, a high energy density of 48.3 Wh kg^−^^1^ was achieved for the assembled supercapacitor at 750 W kg^−^^1^, which is better than other carbon-based supercapacitors that have been reported (Figure 6d). Moreover, a symmetrical supercapacitor can turn on an array of blue LEDs (working voltage of 3 V, inset in Figure 6e), demonstrating the enormous potential for the practical application of h-CPC. The prepared devices also have high cycling stability. Capacitance remained at 92% after 10,000 cycles at 5 A g^−^^1^ (Figure 6e). The Coulombic efficiency is also around 95%, indicating the high electrochemical reversibility of h-CPC based supercapacitor.

## 3. Materials and Methods

### 3.1. Synthesis of Nitrogen-Doped Hierarchical Coal-Based Porous Carbon Materials (h-CPC)

The coal comes from Heishan, Xinjiang, China. In our previous work, the oxidized coal (OC) was pretreated by an oxidation method. Potassium citrate (1.5 g), g-C_3_N_4_ (0.25 g), and OC (0.5 g) were mixed homogenously in a mortar. The mixed powder was then placed in a tube furnace and heated to 700 °C (5 °C min^−1^), and pyrolyzed for 2 h under a N_2_ atmosphere, after which it was allowed to cool naturally to room temperature. The resulting black powder was washed several times with 6 M HCl and deionized water. The resulting carbonaceous material was referred to as h-CPC. The sample obtained by heating potassium citrate (1.5 g) and OC (0.5 g) under the same conditions was referred to as microporous coal-based porous carbon (Mi-CPC) for comparison. In a similar manner, mesoporous coal-based porous carbon (Me-CPC) was prepared by heating g-C_3_N_4_ (0.25 g) and OC (0.5 g).

### 3.2. Characterizations

Morphological characterization was carried out using scanning electron microscopy (SEM, Hitachi, S-4800, Kyoto, Japan) and transmission electron microscopy (TEM, JEOL, JEM-2100F, Kyoto, Japan). X-ray diffraction (XRD, Bruker D8 Advance, Saarbrucken, Germany) was used for phase analysis. A Raman spectrometer (SENTERRA, Bruker, Saarbrucken, Germany) was used to record the spectra. Nitrogen physisorption was conducted at 77 K (Micromeritics, ASAP 2020, Atlanta, CA, USA). X-ray photoelectron spectroscopy (XPS, Thermo, ESCALAB 250Xi, Waltham, MA, USA) was used to characterize functional groups on the surface of samples.

### 3.3. Electrochemical Measurements

An electrochemical workstation (CHI 660E, Chenhua, Shanghai, China) was used to test the electrochemical properties of all samples. The electrode material was composed of active material, acetylene carbon black, and polyvinylidene fluoride with a mass ratio = 8:1:1, which were mixed homogeneously in ethanol to form a homogenous slurry. Then the slurry was cut into square pieces (mass loading for active materials, 2 mg cm^−2^) and then pressed against a nickel foam collector. A total of 6 M KOH was used as the electrolyte in this three-electrode system, Hg/HgO as the reference electrode and Pt plate as the counter electrode. To assemble a symmetrical two-electrode supercapacitor, two identical working electrodes were separated into different electrolytes using a cellulose acetate membrane.

The specific capacitances (*C_s_*, F g^−1^) in the three-electrode system were calculated from GCD curves with Equation (3):*C_s_* = *I* Δ*t*/(*m* Δ*V*)(3)
where *I*/*m* (A/g), Δ*t* (s), and Δ*V* (V) are the discharge current density, discharge time and the potential window, respectively.

In the two-electrode system, the specific capacitance based on a single electrode was calculated by Equation (4)
*C* = 2 *I* Δ*t*/(*m* Δ*V*)(4)
where Δ*V* (V) is the working voltage and *m* (g) represents the mass of active material on a single electrode.

The energy densities and power densities of the symmetric supercapacitors were calculated by Equations (5) and (6).
*E* = *C* (Δ*V*)^2^/28.8(5)
*P* = 3600 *E*/Δ*t*.(6)

## 4. Conclusions

In summary, we successfully synthesized hierarchical porous carbons doped with nitrogen using a green and facile dual template strategy. The resulting h-CPC has high electrical conductivity, good wettability, abundant channels, a large specific surface area, and a high surface N content. The synergy of these properties provides sufficient charge storage space and ultra-fast charge transfer channels. As a result, the h-CPC electrode exhibits excellent energy storage capacity with long-term cycling stability and excellent rate performance. The assembled supercapacitors in the KOH electrolyte and EMIM BF_4_ electrolyte achieved maximum energy densities of 8.3 Wh kg^−^^1^ and 48.3 Wh kg^−^^1^ at power densities of 250 W kg^−^^1^ and 750 W kg^−^^1^, respectively, which are superior to most reported carbon-based supercapacitors. Coupled with abundance and the low-cost of coal resources, this work provides a new approach to the development of advanced carbon materials and opens a new avenue for the practical application of future energy storage technology.

## Figures and Tables

**Figure 1 molecules-28-03660-f001:**
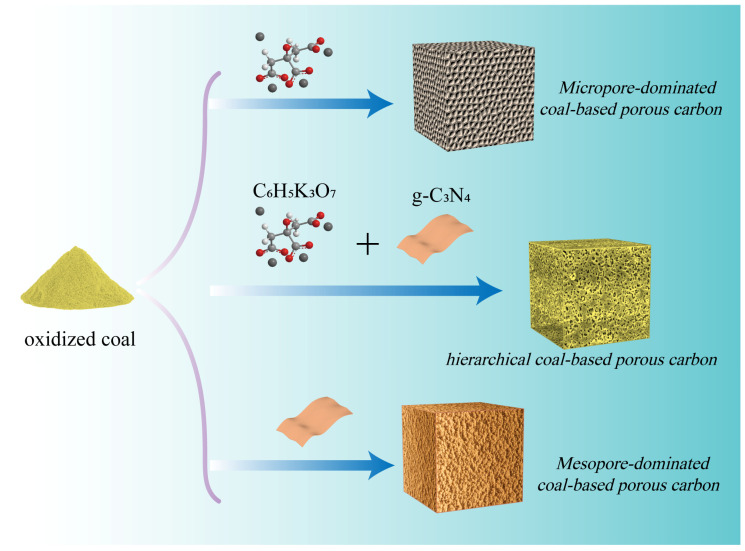
The synthesis scheme of Mi-CPC, Me-CPC, and h-CPC.

**Figure 2 molecules-28-03660-f002:**
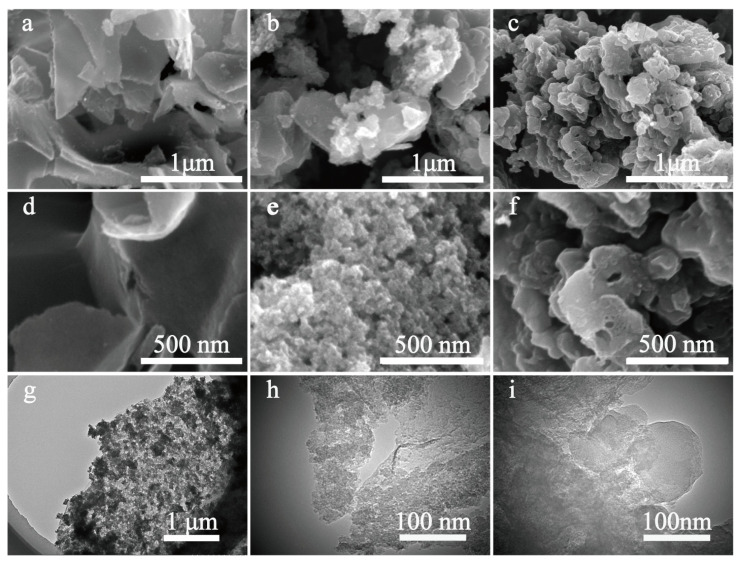
SEM images of the obtained sample: (**a**,**d**) Mi-CPC; (**b**,**e**) h-CPC; (**c**,**f**) Me-CPC; (**g**–**i**) TEM images of h-CPC.

**Figure 3 molecules-28-03660-f003:**
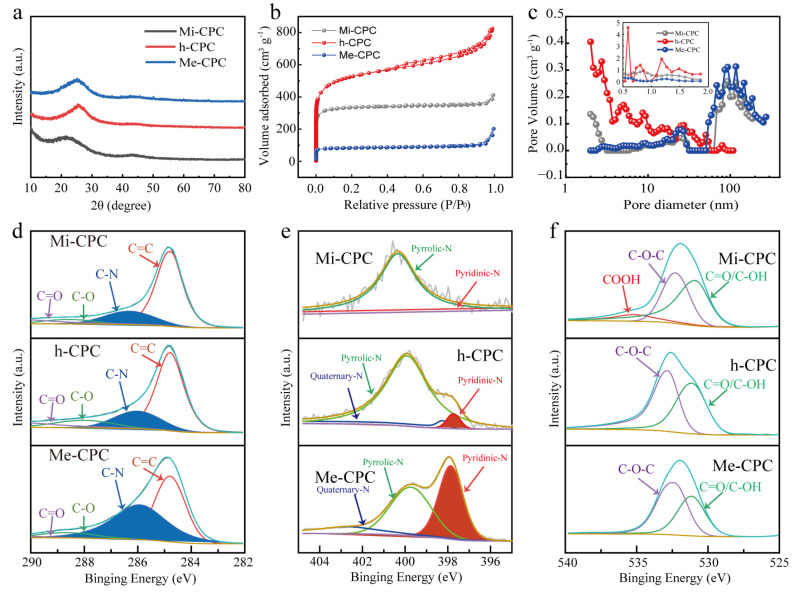
(**a**) XRD, (**b**) N_2_ desorption/adsorption isotherms, and (**c**) pore size distributions of Mi-CPC, h-CPC, and Me-CPC. High-resolution spectra: (**d**) C 1s, (**e**) N 1s, and (**f**) O 1s.

**Figure 4 molecules-28-03660-f004:**
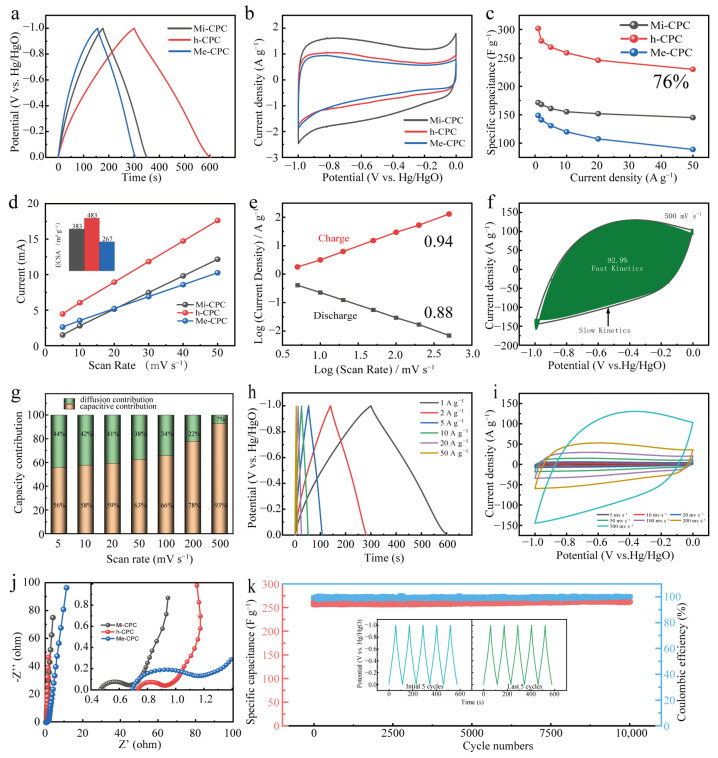
Electrochemical properties of all samples in a three-electrode system: (**a**) GCD curves, (**b**) CV curves, (**c**) specific capacitances at different current densities, (**d**) ECSA, (**e**) I-ν plots, (**f**) contribution of capacitances at 500 mV s^−^^1^, (**g**) capacitance contributions at different scan rates, (**h**) GCD curves at different current densities of h-CPC, (**i**) CV curves at various scan rates, (**j**) Nyquist plots, and (**k**) cycling performance of h-CPC.

**Figure 5 molecules-28-03660-f005:**
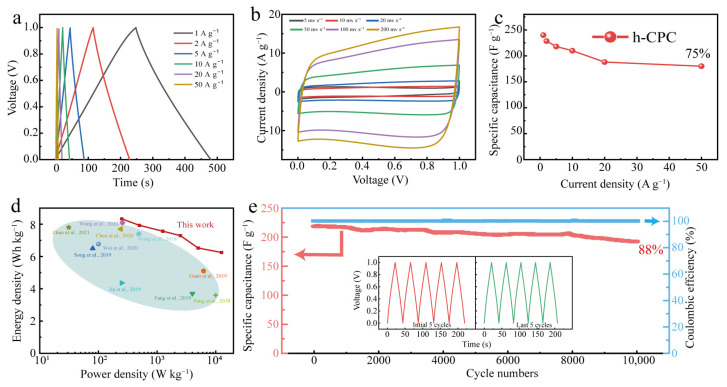
Electrochemical performance of h-CPC in a 6 M KOH for a two-electrode system: (**a**) GCD curves, (**b**) CV curves, (**c**) rate performance, (**d**) Ragone plots [7,12,24,27,32,33,35,36,37,42], and (**e**) cycling performance.

**Figure 6 molecules-28-03660-f006:**
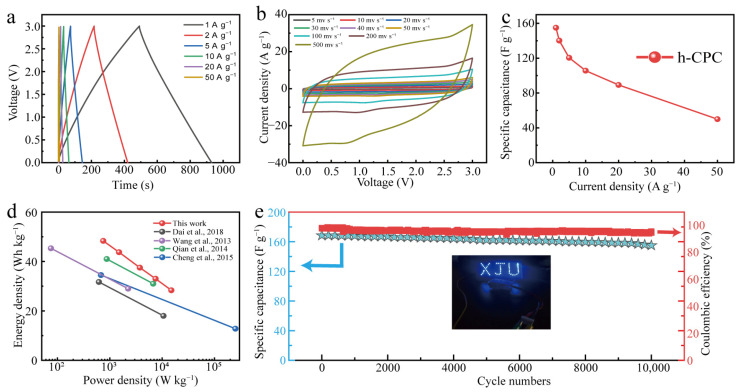
Electrochemical performance of h-CPC in EMIM BF_4_ electrolyte. (**a**) GCD curves, (**b**) CV curves, (**c**) specific capacitance at different current densities, (**d**) Ragone plots [11,18,19,20], and (**e**) cycling performance.

## Data Availability

Data is contained within the article.

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
