# Peer review of "Nitrogen-Doped Hierarchical Porous Carbon Derived from Coal for High-Performance Supercapacitor"

_molecules, 2023, doi:10.3390/molecules28093660_

Round 1

Reviewer 1 Report

In this paper, the authors present a novel approach for preparing coal-based hierarchical porous carbon with nitrogen doping for use in supercapacitors. While the study demonstrates promising results with high specific capacitance, energy density, and cycling stability, there are a few areas that could benefit from further elaboration or exploration. It is recommended that the following issues be addressed and then considered for publication.

1. A more detailed discussion of the dual template strategy and its novelty compared to existing methods would help to better contextualize the work. The paper could benefit from a more comprehensive investigation of the role of nitrogen doping in enhancing the electrochemical performance of the porous carbon. While the authors highlight the cost-effectiveness and simplicity of their approach, a clearer comparison to other carbon-based materials and their respective synthesis methods would strengthen the argument for the practical application of this coal-based hierarchical porous carbon.

2. The layered porous structure does facilitate improved ion transport kinetics. However, volume-to-capacitance is currently a more important metric for evaluating electrode materials. What is the volumetric performance of this nitrogen-doped hierarchical porous carbon?

3. The authors should have introduced the MXene material in the background section because it is a more potential electrode material with the high power density of supercapacitors and the high energy density of batteries. Quality literature that can be referred to is as follows: Adv. Funct. Mater. 2017, 1701264; Matter 2022, 5, 1042-1055; ACS Appl. Energy Mater. 2020, 3, 586-596.

4. The pictures in the article are not clear enough (for all figs) and not beautiful enough (such as Fig. 1, in Fig. 1, the color matching in Fig. 1 is not good enough). The author should improve them.

5. Equations (1)- (6) should be given in the main text.

Reviewer 2 Report

1- The quality of the images is very bad. Replace with better resolution.
2- Why did you choose 6 M KOH electrolyte?
3- I suggest you add EIS analysis from the samples to the article.
4- Please provide an elemental analysis of (XRF - EDS) coal and other synthesized products. The amount of carbon, oxygen, and other elements should be known.
5- Compare the results of your research with similar research (In table).
6- Coal characterization is very important what mineral is coal extracted from?
7- Synthesis is confusing. Amounts of coal and products produced in grams should be mentioned.
8- ECSA results should be included in the main text of the article.
9- What formula did you use to calculate specific capacitance?
10- specific capacitance should be calculated from both CV and GCD values and these values should be compared.

Round 2

Reviewer 1 Report

I have carefully reviewed the authors' previous version of the manuscript and the authors have done a good job of addressing the issues I have raised. I suggest that it is now acceptable and ready for publication.